# The Effect of Time Display Format on Cognitive Performance of Integrated Meteorological Radar Information

**DOI:** 10.3390/bs14090847

**Published:** 2024-09-20

**Authors:** Bo Liu, Yunhe Wang, Yongxin Li

**Affiliations:** 1Institute of Psychology and Behavior, Henan University, Kaifeng 475000, China; liubo2023@henu.edu.cn; 2Department of Preschool Education, Kaifeng Vocational College of Culture and Arts, Kaifeng 475000, China; 104753180351@vip.henu.edu.cn

**Keywords:** cognitive performance, time display format, delay time, time pressure, integrated meteorological radar

## Abstract

A proper time display format is essential for pilots to understand integrated meteorological radar information, thereby making informed flying decisions and steering clear of hazardous weather. Previous studies on time display format supported the advantages of digital format, while some studies found that analog clock format is superior to digital format. This study explored the effect of time display format on the cognitive performance of integrated meteorological radar information through two experiments. Experiment 1 first examined the effects of digital and analog clock displays on the timing of individual processing advance or delay changes in a general scenario. Then, Experiment 2 was conducted in a simulated flight scenario to investigate the advantages and disadvantages of digital and analog clock display in delay time processing with and without time pressure. The results showed the following: (1) Analog clock has more advantages than digital display format in processing the varying time difference. (2) Whether with or without time pressure, analog clock is more conducive to individual cognition of integrated meteorological radar information than digital time display. (3) The length of delay time is an important factor affecting individual time cognition, and it can also affect the cognition of radar information. The longer the delay time, the more difficult it is to identify the time and understand the information. These findings provide a certain reference for the design of the integrated meteorological radar information display interface.

## 1. Introduction

In recent years, short-term and real-time weather forecasting based on data-link images has been widely used in the cockpit of various aircraft to help pilots avoid hazardous weather [1]. The data-link integrated meteorological image is transmitted to the cockpit after integrating, managing, and processing radar information from multiple ground base stations, so the pilot receives time-delayed weather information [2]. Delayed information can cause pilots to misinterpret the true weather conditions and impair their situational awareness, potentially endangering the safety of pilots and aircraft operating near severe weather systems [3]. Studies have shown that pilots tend to overlook the latency of integrated meteorological images or underestimate the significant impact of the uncertainty introduced by the delay on avoidance of hazardous weather [4] and navigation decisions [5]. The reasonable time display format can reflect the delay time of weather information in a simple and understandable way, which helps to reduce the cognitive load of pilots [4]. Therefore, the appropriate design of meteorological radar time display is of great significance to enhance pilots’ cognitive effectiveness, promote pilots to make correct weather decisions, and improve aviation safety.

At present, there are few studies on the time display format of meteorological radar products, and no unified design standard has been formed. Currently, the meteorological radar information interface of commercial aircraft mostly uses digital time display [3]. For example, the Weather Services International (WSI) system communicates time information in a “minute–second” digital format. At the same time, the length of the delay time is distinguished by different colors: green means the delay time is less than five minutes, yellow means five to ten minutes, and more than ten minutes is indicated by orange-red. Digital display is concise and easy to understand, and has been widely used in various product designs [6,7]. However, some studies have shown that digital time displays are not an absolute advantage. For instance, Paivio’s [8] study found that if the operation task involved spatial processing, the individual response to the analog clock would be significantly shorter than that of the digital display. In addition, when it comes to derivative information processing [8], or when the display information changes rapidly [9], analog clocks also exhibit significant advantages over digital time displays.

The design of time display format should be based on individual time perception. The abstractness of time determines the metaphorical nature of time perception [10]. Spatial metaphor is an important pattern for people to perceive time, and the spatial patterns evoked by it can provide a reference frame for individuals to organize time information [11,12]. The digital time display format has a higher degree of fit with the spatial structure of linear mental representation. Lai et al. [10] found that the metaphorical pattern of specific moments is dial simulation. Moreover, individuals adopt a mental rotation approach to perceive temporal changes. This study suggests that analog clock displays can be used for individual perception of time changes. However, in the integrated meteorological radar information display, the pilot must not only sense the moment of change, but also calculate the time difference to confirm the delay time of weather information. However, when individuals need to process the difference in varying time, the advantages and disadvantages of digital and analog clock displays are still unknown.

Hua et al. [3] studied the time display format of meteorological radar images, and the subjects only needed to find the one with the earlier time among the two radar images; the results showed that digital display was more advantageous. However, in a real situation, the pilot has to both confirm the weather image delay time and estimate the plane’s distance from the eye of the storm. Delayed weather image information includes time and space dimensions, and pilots need to establish the corresponding temporal and spatial awareness to maintain good weather situational awareness [13]. The time display information of radar interface can help the pilot establish time awareness. At the same time, the pilot must determine the relative distance between the aircraft and the eye of the storm by combining the delay time, the storm movement speed, and the flight speed, thereby establishing spatial awareness [14,15]. So, when the pilot is in complex and changeable weather conditions that require multiple information processing, which is the better between digital and analog clock display formats?

In addition, time information profoundly affects pilots’ navigation decisions, and even the minimal time delay can threaten aviation safety. The faster and more accurate the delay time perception of meteorological images, the lower is the cognitive load for pilots to make the correct decisions [16]. Therefore, the length of delay time may also affect flight decisions. Furthermore, pilots are often required to make decisions in rapidly changing weather scenarios, and the time pressure cannot be ignored. The greater the time pressure, the greater is the cognitive load on individuals for the same information; too high cognitive load may lead to reduced pilot performance or decision-making errors and even lead to aviation accidents [17]. The appropriate time display format can effectively convey the information required by pilots, reduce their cognitive load, and thus ensure reasonable decisions [18].

Previous research showed that individuals perceive the moments of change in a mental rotation, similar to a dial. So, will analog clocks be better than digital display formats in processing the difference in time of change? Furthermore, when faced with multiple information processing tasks, and with the involvement of delay time length and time pressure, which one is better displayed by digital and analog clocks?

To explore these issues, two experiments were designed in this study. Experiment 1 aimed to explore the effects of digital and analog clock displays on individual processing of varying time differences in a general scenario. The experimental hypothesis is that under the analog clock display format, individuals can process the varying time differences faster and more accurately.

Then, we conducted Experiment 2 with reference to the study of Hua et al. [3]. In Experiment 2, participants were asked to estimate the true distance of the aircraft from the eye of the storm based on simulated meteorological images. Based on the response time and distance deviation values of the subjects, we compared the benefits and drawbacks of digital and analog clock display formats. The experimental hypothesis was that under the simulated clock display format, the response time of the subjects would be shorter and the distance deviation value smaller.

## 2. Experiment 1 Effect of Time Display Format on Individual Processing Change Time Difference

Experiment 1 was conducted in a general scenario. Given that individuals will experience both advanced and delayed time in their daily lives, two levels of advance and delay are set in the direction of time change to explore the influence of digital and analog clock display on individual processing of varying time difference.

### 2.1. Method

#### 2.1.1. Participants

Referring to previous studies on meteorological radar information cognition [3,13,19] and using G*power 3.1 analysis software [20], the medium effect size of 0.25 and the efficacy value of 0.8 were set, and the minimum sample size of this experiment was calculated to be 13 people.

We recruited participants by distributing leaflets at the entrance of the cafeteria, downstairs of the teaching building, and downstairs of the dormitory, and finally recruited a total of 27 college students (12 males), aged 19.10 ± 1.13 years old. They were all right-handed, with normal visual acuity or corrected vision, and no color blindness. A simple dial time recognition test was conducted on the recruited college students using a self-made clock chart containing ten different time points (partially shown in Figure 1). The result showed that all the subjects could read the time quickly and accurately.

#### 2.1.2. Research Design

A three factor within-subjects design of 2 (time display format: digital/analog clock) × 2 (change direction: advance/delay) × 3 (time interval: 1~10 min/11~20 min/21~30 min) was adopted. Referring to Lai et al. [10], we selected a time within 30 min based on rotation within 180 degrees. Moreover, according to the practice that the rotation angle of 20 min is just twice as large as that of 10 min, the variable time is divided into two levels of 10 min and 20 min. In addition, we considered the actual delay time range of current meteorological radar products ranging from 5 min to more than 20 min [3,19]. We selected the change time within the 30 min range and divided it into three levels: 1~10 min, 11~20 min, and 21~30 min. Each variable time included 10 intervals, such as 1~10 min, with intervals ranging from 1 min to 10 min.

Time display format, change direction, and time interval are all within-subject variables. The research aimed to explore the main effects of each independent variable and the interaction effects between each variable. The dependent variable is the response time and the accuracy of identifying the direction of time change, as well as the accuracy of calculating the time differences.

#### 2.1.3. Materials and Apparatus

Adobe Photoshop CS4 drawing software (v.11) was used to create digital and analog clocks displaying time information at different varying times. In reference to the Universal Time Coordinated (UTC) format, we designed the digital time display in the “hour–minute” form (font Arial 14). Based on simple and easy to understand principles and combined with the style of circular design in aircraft cockpit instrument information display [21], we designed a simple analog clock with an hour scale for the internal number of the dial (font Arial 14) and a minute scale for the external number (font Arial 12). All analog clocks looked and function the same, with two longer hands pointing to “minutes” and one shorter hand pointing to “hours”. The width of the digital display was the same as the diameter of the analog clock, both of which were 4 cm (see Figure 2).

E-prime2.0 software was adopted to compile the experimental program. The stimuli were presented by a Panasonic CF-53 computer with a 14-inch screen, 1366 × 768 pixels resolution, and a refresh rate of 60 Hz, while the distance between the subject’s head and the computer screen was approximately 60 cm. The lighting conditions were close to natural light, with an illumination of 120 lx. The entire experimental process had to be kept quiet to avoid interference from irrelevant noise.

#### 2.1.4. Procedure

The experimental procedure consisted of practice and formal experiments. First, the instruction was presented, and after the subjects had fully understood it, they proceeded to the practice section for a total of 24 trials. Then the variable time was presented, and the subjects were asked to judge the direction of the variable time. The direction of change was to press the “-” key in advance and the “+” key in delay. After pressing the button, the time information disappeared, and the subject had to enter the time difference immediately. Then they proceeded to the next trial. In order to balance the experimental sequence, half of the subjects first presented the digital display time, took a rest for 5 s after presenting all of them, and then presented the analog clock display time. The other half of the subjects were presented in the reverse order, with all other settings unchanged. The time information in different display groups was randomly presented, and each subject completed a total of 2 × 2 × 3 × 10 = 120 trials, lasting approximately 25 min. The flowchart of Experiment 1 is shown in Figure 3.

### 2.2. Results

The data were sorted according to the accuracy of identifying the direction of time change and the response time, and the extreme data exceeding the average three standard deviations were eliminated. Three subjects were deleted due to excessive extreme data (more than 20%), and finally 24 subjects (11 males) were included in the analysis. Repeated measurement ANOVA was performed on the data, and the results showed that the main effect and interaction effect of each variable were not significant in terms of the accuracy of identifying the direction of time change (*ps* > 0.05). The response time (RT) of the subjects to identify the direction of time change and the accuracy (ACC) of calculating the time difference under different experimental conditions are shown in Table 1.

#### 2.2.1. Response Time of Identifying the Direction of Time Change

The main effect of time display format was significant, *F*(1, 23) = 32.66, *p* < 0.001, *η*^2^ = 0.59, and the response time of the analog clock (3804.65 ms) was less than that of the digital display (5921.69 ms).

The main effect of the change direction was significant, *F*(1, 23) = 6.11, *p* < 0.05, *η*^2^ = 0.21, and the response time of delay direction (4704.19 ms) was less than the advance direction (5022.15 ms).

The main effect of the time interval was significant, *F*(2, 22) = 104.32, *p* < 0.001, *η*^2^ = 0.91, 1~10 min, and the response time (3595.95 ms) was smaller than other time intervals. Time of 11~20 min (5314.39 ms) is less than 21~30 min response time (5679.18 ms).

The interaction effect between the time change direction and the time interval was significant, *F*(2, 22) = 19.16, *p* < 0.001, *η*^2^ = 0.64. Simple effect analysis showed that the three time intervals were significantly different in the direction of delay (*ps* < 0.05). In the advance direction, the reaction time of 1~10 min interval was significantly smaller than that of other time intervals (*ps* < 0.001). No other effects were significant (*ps* > 0.05).

#### 2.2.2. Accuracy of Calculating the Time Differences

The main effect of the time display format was significant, *F*(1, 23) = 25.19, *p* < 0.001, *η*^2^ = 0.52; the analog clock accuracy (0.97) was higher than the digital display (0.91).

The main effect of time change direction was marginally significant, *F*(1, 23) = 5.76, *p* = 0.052.

The main effect of the time interval was significant, *F*(2, 22) = 12.43, *p* <. 001, *η*^2^ = 0.53, The accuracy of 21~30 min (0.92) was significantly lower than that of the other two time intervals (*ps* < 0.01).

The interaction effect between the change direction and time interval was significant, *F*(2, 22) = 6.53, *p* < 0.01, *η*^2^ = 0.37. Simple effect analysis showed that the accuracy of a 1~10 min interval was significantly higher than the other two time intervals in the direction of delay (*ps* < 0.01); in the direction of advance, there was no significant difference among the three time intervals (*ps* > 0.05).

The interaction effect between the time change direction and the time display format was significant, *F*(1, 23) = 20.01, *p* < 0.001, *η*^2^ = 0.47. Simple effect analysis showed that in the direction of delay, the accuracy of the analog clock was significantly higher than that of the digital display (*p* < 0.001), but there was no significant difference between time display formats in the advance direction (*p* > 0.05). No other effects were significant (*ps* > 0.05).

### 2.3. Summary

Experiment 1 found that there was no significant difference in the accuracy of identifying the direction of time change, indicating that all subjects could accurately identify the time change direction. However, the response time of the subjects in the advance direction of the time change was longer than that in the delayed direction. Additionally, there were significant differences in the response time of the subjects to identify the change direction and the accuracy of calculating the difference in the time display format. Under analog clock display, individuals exhibited shorter response time and higher accuracy. Experiment 1 also found that the longer the time interval, the longer was the response time of the subjects, and the lower was the accuracy.

In addition, one possible problem with Experiment 1 is that when calculating the time difference, the digital format requires subtraction operations, while the analog clock format only requires counting the grid. To rule out the possibility of this affecting the experimental results, we conducted an additional experiment. Specifically, the calculation of the time difference on analog clock display format was changed to compare the difference between the time of the two analog clocks. Then the advantages and disadvantages of digital and analog clock displays were compared in calculating the time difference. The results showed no significant difference from the results of this experiment (*ps* > 0.05), indicating that the results of Experiment 1 are credible.

## 3. Experiment 2 Effect of Time Display Format on Individual Cognition of Simulated Weather Radar Information

In Experiment 1, it was found that the analog clock display format had advantages in the cognitive processing of changing moments in the general situation (time is advanced or delayed). However, the time information of meteorological radar is only delayed. So, will the advantage of analog clocks continue to exist in this specific situation? In addition, how will time pressure and delay time, as inevitable issues in flight scenarios, affect individual cognitive performance of meteorological radar information? To explore these questions, we conducted Experiment 2.

### 3.1. Method

#### 3.1.1. Participants

Referring to previous studies on meteorological radar information cognition [3,13,19] and using G*power 3.1 analysis software [20], the medium effect size of 0.25 and the efficacy value of 0.8 were set, and the minimum sample size of this experiment was calculated to be 13 people.

We recruited participants by distributing leaflets at the entrance of the cafeteria, downstairs of the teaching building, and downstairs of the dormitory. Finally, a total of 48 college students were recruited and divided into two groups: 24 subjects (12 males) in time pressure group, aged 20.05 ± 1.30 years old; 24 subjects (10 males) in no time pressure group, aged 20.05 ± 1.28 years old. All subjects were right-handed, had normal vision or corrected vision, and had no color blindness. The time recognition test was carried out on all subjects as in Experiment 1, and the results showed that all subjects could recognize time quickly and accurately.

#### 3.1.2. Research Design

The experimental designs with or without time pressure were all two factor within-subjects design of 2 (time display format: digital/analog clock) × 3 (delay time: short/medium/long). The short delay time was 1~10 min, the medium delay time was 11~20 min, and the long delay time was 21~30 min.

Time display format and delay time are both within-subject variables. The research aimed to explore the main effects of each independent variable and the interaction effects between each variable. The dependent variables are the response time and distance deviation value of the participants in judging the distance between the aircraft and the eye of the storm.

#### 3.1.3. Materials and Apparatus

Using Adobe Photoshop CS4 drawing software to create simulated meteorological radar images and based on the graph used in Hua et al.’s [3] study, time information in different display formats, as well as information on aircraft position and flight speed, storm position, and moving speed, and observation distance between the two, were added. These pieces of information were located on the right side of the meteorological image, presented in font size 18 in Song typeface, and were 1 cm away from the top and bottom of the image. The real distance between the current aircraft and the eye of the storm can be calculated by using delay time, flight speed, storm moving speed, and observation distance. In all images, the aircraft is moving towards the storm (see Figure 4). Due to the wide variation range of illumination in the light environment of the aircraft cockpit, there is no regulation on the light intensity of the lighting [21]. According to the illumination standard in “Lighting in Indoor Workplaces” (the brightness of the cockpit overhead light is adjustable), the illumination of the place with its own light was 30~50 lx; considering the impact of the aircraft cockpit instrument light, the illumination was set at 40 lx [22]. Experimental apparatus was the same as Experiment 1.

#### 3.1.4. Procedure

The experiment included two parts: no time pressure and time pressure. When there was no time pressure, first, the fixation point “+” 800 ms was presented in the center of the screen. Subsequently, a meteorological radar image was presented on the screen. The task of the subjects was to observe the weather radar map and calculate the real distance between the aircraft and the eye of the storm according to the information in the diagram. After the calculation was completed, the “Enter” key was pressed to enter the distance value input interface. After pressing the key, the distance value had to be entered immediately and then the next trial entered. In order to balance the sequence of the experiments, half of the subjects were presented with digital time display meteorological radar image. First, after all the images were presented, they rested for 5 s, and then were presented with the analog clock display meteorological radar map. The other half were presented in the reverse order, and the rest was completely identical. The images in the same time display format were presented randomly, and each subject completed a total of 3 × 10 × 2 = 60 trials.

When there was time pressure, the presentation time of the meteorological radar images was shortened to 50% of the average response time of the participants under no time pressure. The remaining settings were the same as no time pressure.

Experiment 2 lasted about 30 min, and the formal experiment flow chart is shown in Figure 5.

### 3.2. Results

The experimental data were organized and the distance deviation value of the subjects calculated. The data were screened according to the response time, and the extreme data exceeding the mean plus or minus three standard deviations were eliminated. Finally, data from a total of 48 participants were included in the analysis (24 each with and without time pressure). The response time and distance deviation values judged by subjects with or without time pressure are shown in Table 2.

#### 3.2.1. Response Time

**When there was time pressure.** The main effect of the time display format was significant, *F*(1, 23) = 8.56, *p* < 0.01, *η*^2^ = 0.27; the response time under analog clock display (6737.99 ms) was smaller than that under digital display (7698.83 ms).

The main effect of the delay time was significant, *F*(2, 22) = 74.70, *p* < 0.001, *η*^2^ = 0.77; the response time with short delay (4131.92 ms) was smaller than the other two delay lengths, and the response time with medium delay (7140.10 ms) was smaller than that with long delay (10,383.21 ms).

The interaction effect between the time display format and the delay time was not significant, *F*(2, 22) = 0.72, *p* = 0.498.

**When there was no time pressure.** The main effect of the time display format was significant, *F*(1, 23) = 15.83, *p* < 0.01, *η*^2^ = 0.41; the response time under analog clock display (8160.79 ms) was less than that under digital display (9819.96 ms).

The main effect of the delay time was significant, *F*(2, 22) = 81.75, *p* < 0.001, *η*^2^ = 0.88; the response time with short delay (5522.00 ms) was smaller than the other two delay lengths, and the response time with medium delay (9755.54 ms) was smaller than the long delay (11,693.58 ms).

The interaction effect between the time display format and the delay time was significant, *F*(2, 22) = 6.95, *p* < 0.01, *η*^2^ = 0.39. Simple effect analysis showed that the response time of the analog clock was significantly smaller than that of the digital display under medium and long time delays (*ps* < 0.05). Under short-term delay, there was a significant edge difference between the two display formats, *F*(1, 23) = 4.16, *p* = 0.053.

#### 3.2.2. Distance Deviation Value

**When there was time pressure.** The main effect of the time display format was significant, *F*(1, 23) = 5.04, *p* < 0.05, *η*^2^ = 0.18; the distance deviation value in the analog clock display format (14.26) was smaller than that in the digital display (21.76).

The main effect of the delay time was significant, *F*(2, 22) = 11.21, *p* < 0.001, *η*^2^ = 0.51; the distance deviation value of short-term delay (4.73) was smaller than the other two delay lengths, and the distance deviation value of medium delay (19.67) was smaller than that of long delay (29.65).

The interaction effect between the time display format and the delay time was not significant, *F*(2, 22) = 0.86, *p* = 0.432.

**When there was no time pressure.** The main effect of the time display format was not significant, *F*(1, 23) = 0.26, *p* = 0.616; the main effect of delay time was not significant, *F*(2, 22) = 3.26, *p* = 0.165; and the interaction effect between time display format and delay time was also not significant, *F*(2, 22) = 0.26, *p* = 0.770.

### 3.3. Summary

Experiment 2 found that there was no significant difference in the deviation value of the distance between the aircraft and the eye of the storm when there was no time pressure. However, the response time was significantly different, and the response time under analog clock display was shorter than that under digital display. Moreover, the longer the delay time, the longer was the response time of the individual. When time pressure was present, the response time and distance deviation of the subjects in the analog clock time display were both smaller than that in the digital display. In addition, the longer the delay time, the longer was the individual response time, and the greater was the distance deviation value.

Similarly, Experiment 2 may have had the same problem as Experiment 1. As in Experiment 1, we also conducted an additional experiment to rule out the potential interference. The results showed no significant difference from the results of the current experiment (*ps* > 0.05), indicating that the results of Experiment 2 are credible.

In summary, the results of both Experiment 1 and Experiment 2 showed that the response time of the subjects was faster under the analog clock display format than that of the digital display. Furthermore, the longer the delay time, the longer was the response time of the subjects.

## 4. Discussion

Two experiments were conducted to explore the effects of different time display formats on the cognitive performance of integrated meteorological radar information. The results of Experiment 1 showed that the analog clock display can promote the individual perception of the varying time and the calculation of the time difference value. The results of Experiment 2 indicated that regardless of time pressure, individuals had a better understanding of meteorological radar maps under analog clock display than that of digital display. In addition, under time pressure, individuals had more accurate distance estimation on simulated clock display radar maps, which further supported the superiority of simulated clock display. The results also showed that with and without time pressure, the longer the delay time, the longer was the response and the greater the distance deviation value.

### 4.1. The Advantages of Analog Clock Display in Processing Time Difference Value

This study found that there were significant differences in the response time of individuals to identify the direction of time change and the accuracy of calculating the time difference value in the time display format. Under simulated clock display, individuals reacted faster and with higher accuracy. This indicates that the analog clock is superior to digital display formats in terms of processing time differences. Individual perception of time is metaphorical, and spatial metaphor is an important mode for people to perceive time [11,12]. Further, individuals represent specific moments in the form of analog dials, and process variable moments in the form of mental rotation [10], all of which are highly similar to clocks. So, the analog clock display will promote the individual’s cognition and calculation of varying time. Furthermore, the simulation clock displays a small grid representing one minute and a large grid representing five minutes, and this regularity can also help individuals calculate time differences more quickly.

In addition, compared to digital displays, analog clock displays have more graphic features and can attract individual attention faster [23,24]. Attention plays an important role in time perception, and many theoretical models of attention emphasize the profound influence of attention on time perception [25]. Treisman and Michel [26] were inspired by mechanical clocks and first proposed the Internal Clock Model. The model consists of two basic modules, a pace-maker and an accumulator, in which the pace-maker emits stable signals at certain time intervals, and the number of these signals is recorded into the accumulator, which becomes the basis for the individual representation of time. The time information recorded by the counter can then enter the comparator and the memory, and the comparator compares the current time information with the time information in the memory. According to the model, the analog clock can facilitate an individual in comparing the current time with the past time, thus completing the calculation of the time difference more quickly.

This study also found that there were significant differences in the accuracy of the response time and the calculation of the time difference in the time interval. The longer the time interval, the longer was the response time and the lower the accuracy. This may be because the longer the time interval, the more difficult the task will be, and the greater the psychological load generated by individuals in time perception and mental calculation [27], resulting in delayed response and reduced performance.

### 4.2. The Advantages of Analog Clock Display in Simulating Weather Radar Information Cognition

This study found that under no time pressure, there was no difference in radar information cognitive performance between analog clock display and digital display, but there was a significant difference in response time between the two display formats, while the individual response time was shorter under analog clock display. Under time pressure, individuals responded faster to the radar infographics embedded in the analog clock display and had better cognitive performance. This is because when the information is presented for a short time, the accuracy of the individual’s observation of the radar image decreases. In this case, the analog clock is more vivid and profound than the digital display [23,24], which is more likely to attract the attention of individuals, so that individuals can capture key information in meteorological images more quickly and exhibit more accurate distance estimation.

This study supports the superiority of analog clock display in the process of individual cognition of simulated meteorological radar information, which contradicts the conclusion of previous studies that digital display is more advantageous [6,7]. Digital display is more intuitive in the representation of specific moments, and has a higher degree of fit with linear mental representation spatial structure [28], which is more conducive to subjects completing the time recognition task used in previous studies on time format, thus demonstrating the superiority of digital display. However, the integrated meteorological radar information has a delay characteristic, and the individual needs to perceive the changing time rather than the fixed time. Analog clocks can more vividly represent time variability, and individuals will use analog dial mode to process changing moments [10]. Thus, analog clocks demonstrate obvious advantages in conveying time delay information.

This study also found that no matter with or without time pressure, the longer the delay time, the longer was the response time of individuals to radar information cognition, and the lower the accuracy. Combined with the results of Experiment 1, this may be because the longer the time interval, the greater is the cognitive load of individuals in time perception and computation tasks, and the delay time may affect the time cognition of individuals, and thus, affect their understanding of weather images. The longer the delay time, the greater is the difficulty of time cognition, and the difficulty of individual distance calculation will also increase [13]. Therefore, in the integrated meteorological radar information display, the influence of the delay time should also be considered to avoid the response error caused by it.

### 4.3. Limitations

Several limitations should be borne in mind when interpreting the findings of the present study and contemplating future research. First, the subjects in this study are all college students. Although the experimental tasks in this study did not require any specific knowledge, skills, or experience in dealing with bad weather in flight scenarios, there are still certain differences between college students and pilots. Future studies should be conducted with pilots as subjects to validate and generalize the conclusions of this study. Second, this study used simulated meteorological radar images for experiments, and future research could be carried out by using a dynamic radar instrument interface in a simulated cockpit [13]. Third, the experimental task of this study was relatively singular, whereas a pilot needs to perform multiple task operations in an actual flight [17]. Therefore, future research could adopt a multi-task combination method to create a complex task environment and explore the influence of the time display format on radar information cognition.

## 5. Conclusions

In conclusion, the analog clock display format has advantages in processing the difference in the time of change. Furthermore, no matter with or without time pressure, compared with the digital time display format, the analog clock time display promotes the individual’s cognitive performance on integrated meteorological radar information. In addition, delay time is an important factor affecting individuals’ cognition of meteorological radar information. The longer the delay, the worse is the information cognitive performance. These findings suggested that the analog clock format could be used in the time display design of the integrated weather radar information interface.

## Figures and Tables

**Figure 1 behavsci-14-00847-f001:**
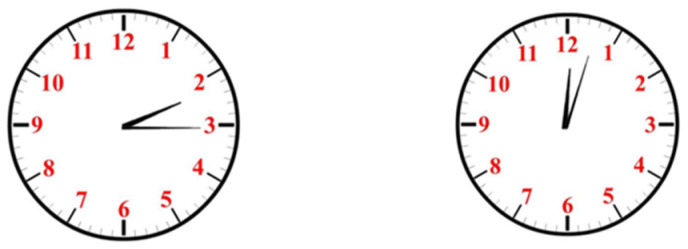
Simple dial time recognition test.

**Figure 2 behavsci-14-00847-f002:**
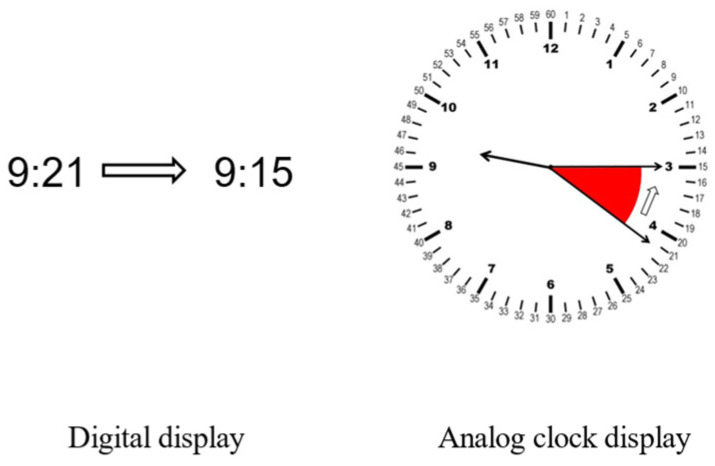
Digital and analog clock time display formats with the same time difference.

**Figure 3 behavsci-14-00847-f003:**
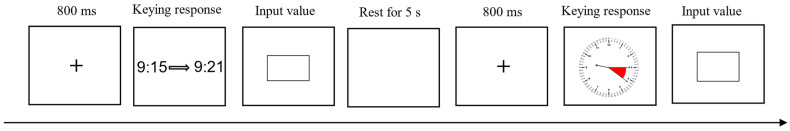
Flowchart of Experiment 1.

**Figure 4 behavsci-14-00847-f004:**
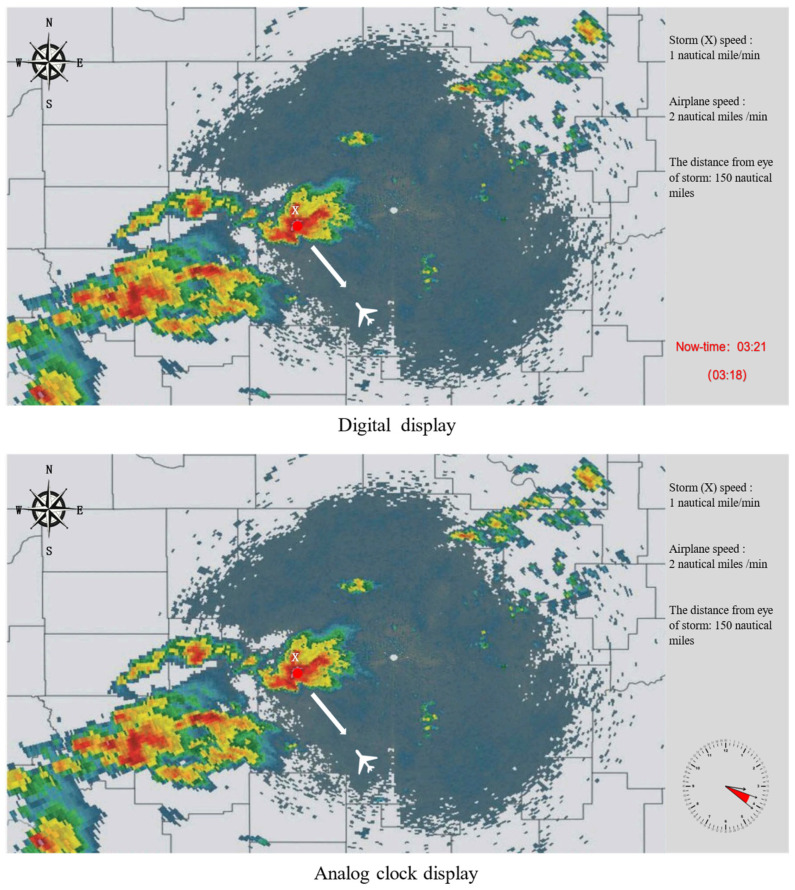
Simulated meteorological radar images with different time display formats. **Note:** The distance of the eye of the storm shown in the figure is from radar observation. The upper hand of the analog clock points to the minute value of the observed time of the radar image, the lower hand points to the minute value of the current time, and the area between the two hands indicates the delay length.

**Figure 5 behavsci-14-00847-f005:**
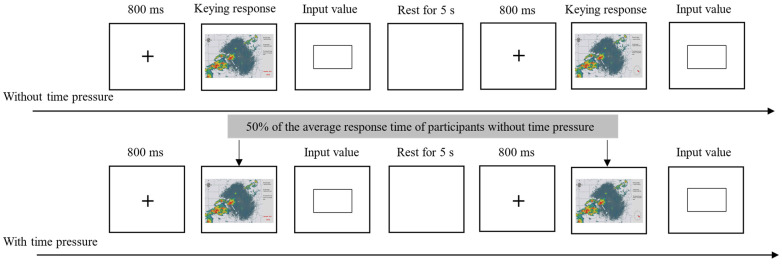
Flowchart of Experiment 2.

**Table 1 behavsci-14-00847-t001:** Descriptive statistical results of participants’ responses under different experimental conditions (*M* ± *SD*).

Time Change Direction	Time Display Format	Time Interval	RT (ms)	ACC
Advance	Digital display	1~10 min	4869.04 ± 260.17	0.93 ± 0.02
		11~20 min	6128.29 ± 339.75	0.96 ± 0.02
		21~30 min	6955.88 ± 359.59	0.94 ± 0.02
	Analog clock display	1~10 min	3208.83 ± 221.93	0.97 ± 0.02
		11~20 min	4155.29 ± 360.02	0.96 ± 0.02
		21~30 min	4815.54 ± 316.24	0.95 ± 0.01
Delay	Digital display	1~10 min	4123.92 ± 309.20	0.95 ± 0.02
		11~20 min	6653.96 ± 312.94	0.88 ± 0.03
		21~30 min	6799.04 ± 333.44	0.83 ± 0.02
	Analog clock display	1~10 min	2182.00 ± 142.32	0.99 ± 0.01
		11~20 min	4320.00 ± 326.22	0.97 ± 0.01
		21~30 min	4146.25 ± 287.76	0.97 ± 0.01

**Table 2 behavsci-14-00847-t002:** Response time and distance deviation values judged by subjects with/without time pressure (*M* ± *SD*).

Time Display Format	Delay Time	Without Time Pressure (24 People)		With Time Pressure (24 People)	
		RT (ms)	Distance Deviation Value (nmi)	RT (ms)	Distance Deviation Value (nmi)
Analog clock display	Short	5134.33 ± 560.21	6.17 ± 2.89	3701.38 ± 403.43	4.42 ± 1.79
	Medium	8847.63 ± 764.85	18.29 ± 13.62	6462.79 ± 652.83	13.08 ± 4.15
	Long	10,500.42 ± 1015.17	15.33 ± 6.51	10,049.79 ± 833.31	25.29 ± 7.71
Digital display	Short	5909.67 ± 377.95	4.25 ± 2.32	4562.46 ± 438.57	5.04 ± 1.69
	Medium	10,663.46 ± 684.18	9.79 ± 4.68	7817.42 ± 676.69	26.25 ± 6.91
	Long	12,886.75 ± 881.67	15.75 ± 4.18	10,716.63 ± 960.83	34.00 ± 8.13

## Data Availability

The data that support the findings of this study are available from the corresponding author upon request.

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
