# Peer review of "The Effect of Time Display Format on Cognitive Performance of Integrated Meteorological Radar Information"

_behavsci, 2024, doi:10.3390/bs14090847_

Round 1
Reviewer 1 Report
Comments and Suggestions for Authors
Pilots must be able to interpret radar data in an appropriate time display format in order to make informed flying decisions and steer clear of dangerous weather. While some studies revealed that analog clock format is preferable to digital format, previous studies on time display format backed the benefits of digital format. Through two studies, this study investigated how the time display format affected the integrated meteorological radar information's cognitive function. First, in a generic scenario, Experiment 1 investigated how digital and analog clock displays affected the timing of individual processor advance or delay adjustments. In order to examine the benefits and drawbacks of digital and analog clock displays in delay time processing with and without time pressure, experiment 2 was then carried out in a flight simulator.
Although the manuscript is in good shape however required following queries to be resolved before the final recommendation:
- Writing experiment 1 before the Methods section is a fairly uncommon approach. Kindly refer to a few published articles regarding the arrangement.
- The p-values must be added to tables where neccesary for better explaination. It is very difficult to find the siginficance levels.
- Which statistical tests, given the low number of participants, were chosen by the authors.
- Whether, some of the outcomes from the two experiments correlated? How much. To elaborate, a separate subsection is required.
- Figures lacks clarity. Please redraw them to explain the purpose in a comprehensible manner and typography.
- IT would be better to add the detailed captions of figures having results.
-It doesn't seem good to separate the two experiments and discuss them in separate methods sections; it would confuse the reader. Please write them in a comprehensive manner at once.
- The conclusion portion needs to be improved. The authors need to confirm that the references are consistent.
Author Response
Dear reviewer,
Thank you very much for giving us the opportunity to revise the manuscript! Thank you very much for the time and effort you put into our manuscript, as well as for your valuable comments. These comments are of great help to the revision and improvement of our paper, and have important guiding significance for our study.
We have carefully studied the opinions put forward by you and revised the paper according to the opinions. The modified part has been marked in blue in the re-submitted manuscript. The comments are replied one by one as follows:
Pilots must be able to interpret radar data in an appropriate time display format in order to make informed flying decisions and steer clear of dangerous weather. While some studies revealed that analog clock format is preferable to digital format, previous studies on time display format backed the benefits of digital format. Through two studies, this study investigated how the time display format affected the integrated meteorological radar information’s cognitive function. First, in a generic scenario, Experiment 1 investigated how digital and analog clock displays affected the timing of individual processor advance or delay adjustments. In order to examine the benefits and drawbacks of digital and analog clock displays in delay time processing with and without time pressure, experiment 2 was then carried out in a flight simulator.
Although the manuscript is in good shape, however required following queries to be resolved before the final recommendation:
Comments 1: Writing experiment 1 before the Methods section is a fairly uncommon approach. Kindly refer to a few published articles regarding the arrangement.
Response 1: Thanks a lot for your valuable comment. Two experiments were conducted in this study. In terms of the arrangement of the writing structure, we treated each experiment as a separate section, with Experiment 1 and Experiment 2 as sections titles, resulting in Experiment 1 being placed before the Methods section. As you mentioned, although this is not a common writing style, it should be considered reasonable in this paper.
Comments 2: The p-values must be added to tables where neccesary for better explaination. It is very difficult to find the siginficance levels.
Response 2: Thank you for your valuable comment. The results of this study are indeed quite complicated, and we have made efforts to present them in a clear manner. Our approach is to present descriptive statistical results for each experimental condition in tabular form, and then present the results of the main effects of each research variable and the interaction effects between variables in several paragraphs. The statistical discrepancies between different experimental conditions are presented in the corresponding paragraphs, so the p-values are not presented in the table. Moreover, we have rearranged the experimental results section and annotated them in italics and bold.
Please refer to sections 2.2.1, 2.2.2, 3.2.1, and 3.2.2, highlighted in blue for the modified content.
Comments 3: Which statistical tests, given the low number of participants, were chosen by the authors.
Response 3: Thank you for pointing this out. We used G-power 3.1 (Faul et al., 2007), to calculate the minimum sample size. The selection and setting of relevant statistical parameters are as follows:
[Test family] = F tests
[Statistical test] = ANOVA: Repeated measures, within factors
[Type of power analysis] = A prior: Compute required sample size – given a, power and effect size.
The a of 0.05, medium effect size of 0.25 and the efficacy value of 0.8 were set.
Please refer to section 2.1.1, lines 114 to 116, highlighted in blue for the modified content.
References
Faul, F., Erdfelder, E., Lang, A. G., & Buchner, A. (2007). G*Power 3: A flexible statistical power analysis program for the social, behavioral, and biomedical sciences. Behavior Research Methods, 39(2), 175-191.
Comments 4: Whether, some of the outcomes from the two experiments correlated? How much. To elaborate, a separate subsection is required.
Response 4: Thank you for pointing this out. In Experiment 1, the advantages of the analog clock display format in the time of processing advance or delay change were found in the general situation. Subsequently, in the flight scenarios, the benefits and drawbacks of digital and analog clock displays in delay time processing with and without time pressure were examined in Experiment 2. The results indicated that whether with or without time pressure, analog clock is more conducive to individual cognition of integrated meteorological radar information than digital time display.
In summary, the results of both Experiment 1 and Experiment 2 showed that the response time of the subjects is faster under the analog clock display format than that of digital display. Furthermore, the longer the delay time, the longer the response time of the subjects.
Please refer to section 3.3, lines 343 to 346, highlighted in blue for the modified content.
Comments 5: Figures lacks clarity. Please redraw them to explain the purpose in a comprehensible manner and typography.
It would be better to add the detailed captions of figures having results.
Response 5: Thanks a lot for your valuable comment. We have rearranged the experimental results section and annotated them in italics and bold. Our arrangement of results follows the sequence from the main effects of each variable to the interaction effects between variables.
Please refer to sections 2.2.1, 2.2.2, 3.2.1, and 3.2.2, highlighted in blue for the modified content.
Comments 6: It doesn’t seem good to separate the two experiments and discuss them in separate methods sections; it would confuse the reader. Please write them in a comprehensive manner at once.
Response 6: Thanks a lot for your valuable comment. We write Experiment 1 and Experiment 2 as two separate sections because they are two different experiments with differences in participants, research design, independent variables and dependent variables, materials, procedures, and results. If they are written together in a comprehensive approach, we are concerned that some aspects about these two experiments that cannot be explained clearly.
Comments 7: The conclusion portion needs to be improved. The authors need to confirm that the references are consistent.
Response 7: Thanks a lot for your valuable comment. We discussed the research results from two aspects: “The advantages of analog clock display in processing time difference value” and “The advantages of analog clock display in simulating weather radar information cognition”. We have added some elements to enrich and deepen the discussion. Please refer to sections 4.1, and 4.2, highlighted in blue for the modified content.
We have also checked the references to make sure they were consistent.
Finally, I would like to express my gratitude once again for your hard work! I hope that with your guidance and assistance, we can complete an excellent paper.
If you have any questions, please do not hesitate to contact us.
Reviewer 2 Report
Comments and Suggestions for Authors
Overall, the article is good. This study explored the impact of time display format on the cognitive performance of integrated meteorological radar information through two experiments, and found the superiority of simulated clock display format. The research conclusions have certain reference significance for the design of integrated meteorological radar information display interface. However, there are still several points that need to be improved. My comments are as follows:
1. Is the appearance/functional design of the simulated clock in the paper consistent with the theme of integrated meteorological radar information cognition in the research? The paper only mentions the reference to previous research on the time display format of meteorological radar (Hua et al., 2019), and it is necessary to clarify the issue of conformity.
2. For the explanation of the number of subjects in the experiment, please refer to the relevant article, which suggests a more reasonable one.
3. Is the setting of the three levels of time interval (1-10min, 11-20min, 21-30min) reasonable in the experiment? It is possible to divide it into three levels, but why such divisions as 1-10min? Is this time interval reasonable?
4. The authors have marked the time difference of the analog clock in red. Is the advantage gained related to the red mark?
5. In this study, college students are used as subjects, but there are great theoretical differences between pilot subjects and college students. Variables such as with and without time pressure have less consistent effects across different subjects. What does the author think about this?
Author Response
Dear reviewer,
Thank you very much for giving us the opportunity to revise the manuscript! Thank you very much for the time and effort you put into our manuscript, as well as for your valuable comments. These comments are of great help to the revision and improvement of our paper, and have important guiding significance for our study.
We have carefully studied the opinions put forward by you and revised the paper according to the opinions. The modified part has been marked in blue in the re-submitted manuscript. The comments are replied one by one as follows:
Overall, the article is good. This study explored the impact of time display format on the cognitive performance of integrated meteorological radar information through two experiments, and found the superiority of simulated clock display format. The research conclusions have certain reference significance for the design of integrated meteorological radar information display interface. However, there are still several points that need to be improved. My comments are as follows:
Comment 1: Is the appearance/functional design of the simulated clock in the paper consistent with the theme of integrated meteorological radar information cognition in the research? The paper only mentions the reference to previous research on the time display format of meteorological radar (Hua et al., 2019), and it is necessary to clarify the issue of conformity.
Response 1: Thank you for pointing this out. The appearance/functional design of the simulated clock in this article is consistent with the theme of integrated meteorological radar information cognition in the study, with two pieces of evidence. Firstly, in the design of aircraft cockpit instrument information display forms, there are common forms of circular form, semi-circular form, open window type, vertical type, horizontal type and so on. Among them, the circular instrument can show the trend of the instrument indication, and its eye movement route is more economical than perceiving linear instrument, and it also conforms to people’s usual habit of watching the instrument (You et al., 2017). Therefore, using a simple simulated clock to display time information in the experiment is in line with the design principles of aircraft cockpit instrument display form. Secondly, there has been a precedent for using a simple analog clock form to conduct research on meteorological radar information cognition (Hua et al., 2019), which has proven that this display form is usable, and valuable conclusions have been obtained on this basis. Therefore, we learned from previous studies and followed this design.
Combining the above two points, it can be considered that the appearance/function design of analog clock production in this paper is suitable for the theme of integrated weather radar information cognition.
References
You,X., Ji, M., & Jiao, W. (2017). Aviation Psychology: Theory, Practice and Application. Zhejiang Education Publishing House.
Hua, L., Ling, C., & Thomas, R. (2019). Timestamp representative of weather radar images in the cockpit. The International Journal of Aerospace Psychology, 29(3-4), 86-97.
Comment 2: For the explanation of the number of subjects in the experiment, please refer to the relevant article, which suggests a more reasonable one.
Response 2: Thanks a lot for your valuable comment. In selecting the number of participants for the two experiments, we referred to previous research on meteorological radar information cognition (Hua et al., 2015; Hua et al., 2019, 2022), and use G-power 3.1 (Faul et al., 2007), with a moderate effect size of 0.25 and an efficacy value of 0.8, to calculate the minimum sample size. Combining the two aspects, we determined the number of participants in experiment 1 and experiment 2.
References
Hua, L., Ling, C., & Thomas, R. (2019). Timestamp representative of weather radar images in the cockpit. The International Journal of Aerospace Psychology, 29(3-4), 86-97.
Hua, L., Ling, C., & Thomas, R. (2022). Effects of delayed weather radar images on pilots’ spatial awareness. Applied Ergonomics, 103598.
Hua, L., Chen, L., & Thomas, R. (2015). Effects of delayed radar information on distance estimation. Proceedings of the Human Factors and Ergonomics Society Annual Meeting, 59(1), 150-154.
Faul, F., Erdfelder, E., Lang, A. G., & Buchner, A. (2007). G*Power 3: A flexible statistical power analysis program for the social, behavioral, and biomedical sciences. Behavior Research Methods, 39(2), 175-191.
Comment 3: Is the setting of the three levels of time interval (1-10min, 11-20min, 21-30min) reasonable in the experiment? It is possible to divide it into three levels, but why such divisions as 1-10min? Is this time interval reasonable?
Response 3: Thank you for pointing this out. In terms of setting the time interval level, we mainly based on two considerations: Firstly, some researchers have pointed out that the delay time of commonly used meteorological radar products ranges from about 5 to more than 20 minutes (Hua et al., 2019, 2022). Secondly, Lai et al. (2014) conducted experiments within 30 minutes based on a rotation within 180 degrees, and found that the metaphorical pattern of specific moments was dial simulation. Moreover, individuals process the moment of change in a mental rotation. Combining the two, we chose a change moment in the 30-minute range in the study. In addition, Lai et al. (2014) pointed out that the rotation angle of 20 minutes was just twice as large as that of 10 minutes, which was convenient for participants to perform mental rotation. Therefore, we follow this practice and set the time interval to three levels of 1-10min, 11-20min and 21-30min.
References
Hua, L., Ling, C., & Thomas, R. (2019). Timestamp representative of weather radar images in the cockpit. The International Journal of Aerospace Psychology, 29(3-4), 86-97.
Hua, L., Ling, C., & Thomas, R. (2022). Effects of delayed weather radar images on pilots’ spatial awareness. Applied Ergonomics, 103598.
Lai, S., Lu, Z., Zhou, M., & He, X. (2014). The Study on the Temporal Metaphor at the Level of Moment: Clock Face Simulation. Journal of Psychological Science, 37(1), 2-9.
Comment 4: The authors have marked the time difference of the analog clock in red. Is the advantage gained related to the red mark?
Response 4: Thank you for pointing this out. We explain the practice of "marking the time difference in red" from the following two aspects.
Firstly, in contrast, digital display is more direct, allowing participants to see two moments at a glance, while analog dials have more complex characteristic attributes (including many pointers, scales, and values), which may cause individual perceptual errors and interfere with experimental results. However, our experiment focuses on the impact of digital and analog clock display on the moment difference of individual processing changes, so the potential impact of individual deviation in cognitive moment on the result should be avoided. Therefore, we use red to mark the minute value pointed by the two long pointers under the analog clock.
Secondly, although the time variation of the analog clock was marked in red, the value of the time difference was not directly given, so the participants cannot directly obtain the time difference and still needed to perform psychological calculations to obtain the time difference. In fact, after the completion of the experiment, we also asked the participants about the way they obtained the time difference, and the participants indicated that even though the time change under the analog clock display was marked in red, they still needed to calculate the time difference according to the scale of the dial. In other words, whether it is a digital display or an analog clock with a red mark, the participants need to perform mental calculations to know the time difference. Then, it can be assumed that the advantage of the clock display has nothing to do with the red marker.
In summary, the time difference is marked in red, and it can be argued that the advantages of analog clock display have nothing to do with this red mark.
Comment 5: In this study, college students are used as subjects, but there are great theoretical differences between pilot subjects and college students. Variables such as with and without time pressure have less consistent effects across different subjects. What does the author think about this?
Response 5: Thanks a lot for your valuable comment. This study designed a simple meteorological radar information cognition task, mainly including time processing and distance estimation. These tasks do not require any specific knowledge, skills and experience in handling severe weather in flight situations, and can be regarded as the universal ability of individuals to a certain extent. Therefore, although college students are selected as participants in this study, the results obtained are still of certain significance.
As you mentioned, selecting pilots as subjects is more representative, and the conclusions obtained may differ from the current results, and also have more pertinence and promotion value. However, due to the current experimental conditions, we only recruited college students to participate in this study. This shortcoming has been pointed out in the research limitations of this paper: future research can select pilots as the object of in-depth study, and design experimental tasks that are closer to real aviation scenarios to further explore the design of meteorological radar time information display format.
Finally, I would like to express my gratitude once again for your hard work! I hope that with your guidance and assistance, we can complete an excellent paper.
If you have any questions, please do not hesitate to contact us.
Reviewer 3 Report
Comments and Suggestions for Authors<The Effect of Time Display Format on Cognitive Performance of Integrated Meteorological Radar Information> aimed to compare the processing of digital and analog clock formats of temporal information, and examines their effects on the cognitive performance of integrated meteorological radar information. The biggest problem is that the digital format requires subtraction operations, while the analog clock format only requires counting grids. The difference between the two is not the difference between the representation of digit and analog clock themselves, but the difference between subtraction operations and counting grids. There are the following problems.
1. Introduction. It is necessary to systematically review the relevant research on the two formats of time presentation, digital and analog clock, and analyze the theoretical basis of these two forms of time presentation.
2. The introduction states that some studies support that the digital format is better than the analog clock format, while other studies support that the analog clock format is better than the digital format. This article supports that the analog clock is better, which is just more evidence to support the analog clock is better, and there is not much innovation. This article should design experiments to prove under what circumstances digital is better and under what circumstances the analog clock is better.
3. The introduction should preferably be a hypothesis-testing style, that is, to propose a clear hypothesis and then design an experiment to prove the hypothesis. The introduction of this article does not have an obvious hypothesis.
4. G*power 3.1 analysis software, the research design needs to be explained, such as within-subject or between-subject, factors, and whether the main effect or interaction is of interest. Readers can repeat the power analysis.
5. "were randomly recruited", Several experiments do not randomly select subjects, such as posting posters and subject registration, which is not random. It is necessary to confirm whether it is truly random.
6. In Study 1, the digital format requires subtracting 9:15 from 9:21, and the analog clock format only requires counting the grids of red areas. These two tasks are not comparable. Obviously, the counting grids in analog clock presentation is simpler. The results of Experiment 1 only mean that the difficulty of counting grids is less than the subtraction operation. The counting of grids in the red area should be changed to comparing the difference between the time of the two analog clock. Experiment 2 also has this problem.
7. “2.3. Summary” simply states the results and should be discussed.
Comments on the Quality of English LanguageThis article should be edited by a professional translation company
Author Response
Dear reviewer,
Thank you very much for giving us the opportunity to revise the manuscript! Thank you very much for the time and effort you put into our manuscript, as well as for your valuable comments. These comments are of great help to the revision and improvement of our paper, and have important guiding significance for our study.
We have carefully studied the opinions put forward by you and revised the paper according to the opinions. The modified part has been marked in blue in the re-submitted manuscript. The comments are replied one by one as follows:
<The Effect of Time Display Format on Cognitive Performance of Integrated Meteorological Radar Information> aimed to compare the processing of digital and analog clock formats of temporal information, and examines their effects on the cognitive performance of integrated meteorological radar information. The biggest problem is that the digital format requires subtraction operations, while the analog clock format only requires counting grids. The difference between the two is not the difference between the representation of digit and analog clock themselves, but the difference between subtraction operations and counting grids. There are the following problems.
Comment 1: Introduction. It is necessary to systematically review the relevant research on the two formats of time presentation, digital and analog clock, and analyze the theoretical basis of these two forms of time presentation.
Response 1: Thanks a lot for your valuable comment. We have systematically reviewed the relevant research on the two formats of digital and analog clock display. Individual perception of time is metaphorical, and spatial metaphor is an important model for people to perceive time. The spatial patterns evoked by spatial metaphor can provide a reference framework for individuals to organize time information. The format of digital time display fits well with the spatial structure of linear mental representation, and people’s metaphorical patterns for specific moments are dial simulations, which is the theoretical basis of digital and analog clock display respectively. Please refer to the blue content in lines 58 to 65 of the manuscript for specific modifications.
Comment 2: The introduction states that some studies support that the digital format is better than the analog clock format, while other studies support that the analog clock format is better than the digital format. This article supports that the analog clock is better, which is just more evidence to support the analog clock is better, and there is not much innovation. This article should design experiments to prove under what circumstances digital is better and under what circumstances the analog clock is better.
Response 2: Thank you for pointing this out. The main purpose of this paper is to explore the impact of digital and analog clock time display formats on pilot’s cognition performance of integrated meteorological radar information, and to explore which one is better. It’s not about finding circumstances in which digital display format is better? Under what circumstances is the analog clock display format better?
Moreover, this study revealed that in simulated flight scenarios, analog clock format is preferable to digital format when faced with multiple information processing tasks, and with the involvement of delay time length and time pressure. This indicates that the main research problems have been addressed in this study.
Comment 3: The introduction should preferably be a hypothesis-testing style, that is, to propose a clear hypothesis and then design an experiment to prove the hypothesis. The introduction of this article does not have an obvious hypothesis.
Response 3: Thanks a lot for your valuable comment. We have added the corresponding experimental hypothesis in the introduction of this paper. Please refer to the blue content in lines 101 to 103 and lines 108 to 110 of the manuscript for specific modifications.
Comment 4: G*power 3.1 analysis software, the research design needs to be explained, such as within-subject or between-subject, factors, and whether the main effect or interaction is of interest. Readers can repeat the power analysis.
Response 4: Thank you for pointing this out.
Experiment 1 adopted a three factor within-subjects design of 2 (time display format: digital/analog clock) × 2 (change direction: advance/delay) × 3 (time interval: 1~10min/11~20min/21~30min). In Experiment 2, the experimental designs with or without time pressure were all two factor within-subjects design of 2 (time display format: digital/analog clock) × 3 (delay time: short/medium/long).
We used G-power 3.1 (Faul et al., 2007), to calculate the minimum sample size. The selection and setting of relevant statistical parameters are as follows:
[Test family] = F tests,
[Statistical test] = ANOVA: Repeated measures, within factors,
[Type of power analysis] = A prior: Compute required sample size – given a, power and effect size.
The a of 0.05, medium effect size of 0.25 and the efficacy value of 0.8 were set.
Please refer to section 2.1.1, lines 114 to 116, and section 3.1.1, lines 235 to 238 highlighted in blue for the detailed content.
References
Faul, F., Erdfelder, E., Lang, A. G., & Buchner, A. (2007). G*Power 3: A flexible statistical power analysis program for the social, behavioral, and biomedical sciences. Behavior Research Methods, 39(2), 175-191.
Comment 5: “were randomly recruited”, Several experiments do not randomly select subjects, such as posting posters and subject registration, which is not random. It is necessary to confirm whether it is truly random.
Response 5: Thank you for pointing this out. We did not recruit participants by posting posters and registering them. We recruited participants by distributing leaflets at the entrance of the cafeteria, downstairs of the teaching building, and downstairs of the dormitory, which largely ensured the randomness of the recruitment process.
Comment 6: In Study 1, the digital format requires subtracting 9:15 from 9:21, and the analog clock format only requires counting the grids of red areas. These two tasks are not comparable. Obviously, the counting grids in analog clock presentation is simpler. The results of Experiment 1 only mean that the difficulty of counting grids is less than the subtraction operation. The counting of grids in the red area should be changed to comparing the difference between the time of the two analog clocks. Experiment 2 also has this problem.
Response 6: Thanks a lot for your valuable comment. You mentioned that in calculating the time difference, the analog clock format only requires counting the grid in the red area, while the digital format requires numerical calculation, so the counting under the analog clock display is simpler. This may be a strategy adopted by the subjects, which may result in learning effects after multiple trials. However, it is also possible that the subjects calculated based on two moments under simulated clock display. In fact, we asked the subjects after they had completed the experiments, and most of the subjects replied that they had calculated the difference in time on the dial. We acknowledge that what you mentioned is a limitation of this study, which we have pointed out in the limitations section and will be further refined in the future study.
Please refer to the blue content in lines 438 to 441 of the manuscript for specific modifications.
Comment 7: “2.3. Summary” simply states the results and should be discussed.
Response 7: Thank you for pointing this out. As you mentioned, in section 2.3, we only summarized the experimental results and did not discuss the results. However, we discussed the experimental results in detail in section 4.1, so we did not discuss them in “2.3. Summary”.
Finally, I would like to express my gratitude once again for your hard work! I hope that with your guidance and assistance, we can complete an excellent paper.
If you have any questions, please do not hesitate to contact us.
Round 2
Reviewer 1 Report
Comments and Suggestions for Authors
The authors complied well to this reviewrs comments that warrants its acceptance.
Author Response
Comment: The authors complied well to this reviewers comments that warrants its acceptance.
Response: Thank you very much for your careful review and hard work on this paper.
Reviewer 3 Report
Comments and Suggestions for Authors
The revised version of this paper has hardly any improvement, and almost all the questions have not been well answered. The correct way to revise the paper should be to do additional experiments and refer to relevant papers to improve the research in this paper.
For example: I mentioned: "The biggest problem is that the digital format requires subtraction operations, while the analog clock format only requires counting grids. The difference between the two is not the difference between the representation of digit and analog clock themselves, but the difference between subtraction operations and counting grids." You should do an additional experiment to prove that the possibility I mentioned does not exist. Instead of using this possibility as a limitation.
I mentioned: "G*power 3.1 analysis software, the research design needs to be explained, such as within-subject or between-subject, factors, and whether the main effect or interaction is of interest. Readers can repeat the power analysis." You should describe the calculation process in the paper.
I mentioned: "were randomly recruited", Several experiments do not randomly select subjects, such as posting posters and subject registration, which is not random. It is necessary to confirm whether it is truly random.", Strict random experiments are to assign numbers to all individuals in the sampling pool, and then use computers to randomly select the sample. Posting posters and handing out flyers are not random sampling. You should delete "randomly recruited" and directly write how to recruit subjects.
I mentioned: "The introduction should preferably be a hypothesis-testing style, that is, to propose a clear hypothesis and then design an experiment to prove the hypothesis. The introduction of this article does not have an obvious hypothesis.", You should propose theoretical hypotheses based on existing literature and theories through reasoning. Instead of simply adding a sentence, we assume that..., without reasoning process and theoretical and empirical basis.
Comments on the Quality of English LanguageThis article needs to be edited by an English translation company
Author Response
Thanks a lot for your careful review and hard work. We are very sorry that we did not respond appropriately to your comments. We have carefully studied your comments and made corresponding modifications in the revised paper and marked them in red. We hope our modification will be satisfactory to you. Your comments are very helpful in improving the quality of our article. The responses are as follows:
Comment 1: I mentioned: “The biggest problem is that the digital format requires subtraction operations, while the analog clock format only requires counting grids. The difference between the two is not the difference between the representation of digit and analog clock themselves, but the difference between subtraction operations and counting grids.” You should do an additional experiment to prove that the possibility I mentioned does not exist. Instead of using this possibility as a limitation.
Response 1: Thanks a lot for your valuable comment. Indeed, as you mentioned, the design of digital and analog clock display formats is not rigorous enough in the calculation of time differences. It is possible that the digital format requires subtraction operations, while the analog clock format only requires counting grids. To rule out this possibility of interfering with the results of the study, we did additional experiments. Specifically, the calculation of the time difference on analog clock display format is changed to compare the difference between the time of the two analog clocks. Then compare the advantages and disadvantages of digital and analog clock displays in calculating the time difference. The results showed no significant difference from the results of this study, indicating that the results of the current study is credible.
Please refer to the red content in 2.3. Summary and 3.3. Summary section of the revised manuscript for specific modifications.
Comment 2: I mentioned: “G*power 3.1 analysis software, the research design needs to be explained, such as within-subject or between-subject, factors, and whether the main effect or interaction is of interest. Readers can repeat the power analysis.” You should describe the calculation process in the paper.
Response 2: Thanks a lot for your valuable comment. We apologize for not responding appropriately to your comment. Based on your recommendation, we have explained the research design in detail. As for the calculation process of power analysis, we have described the relevant parameters in sections 2.1.1 and 3.1.1 of the paper, and also listed the specific operation process in the last reply.
Please refer to the red content in 2.1.2. Research Design and 3.1.2. Research Design section of the revised manuscript for specific modifications.
Comment 3: I mentioned: “were randomly recruited”, Several experiments do not randomly select subjects, such as posting posters and subject registration, which is not random. It is necessary to confirm whether it is truly random.”, Strict random experiments are to assign numbers to all individuals in the sampling pool, and then use computers to randomly select the sample. Posting posters and handing out flyers are not random sampling. You should delete “randomly recruited” and directly write how to recruit subjects.
Response 3: Thanks a lot for your valuable comment. We apologize for not responding appropriately to your comment. We were indeed not rigorous enough in the description of the participant recruitment. According to your recommendation, we have deleted “randomly recruited” and directly wrote the approach of recruiting subjects.
Please refer to the red content in 2.1.1. Participants and 3.1.1. Participants section of the revised manuscript for specific modifications.
Comment 4: I mentioned: “The introduction should preferably be a hypothesis-testing style, that is, to propose a clear hypothesis and then design an experiment to prove the hypothesis. The introduction of this article does not have an obvious hypothesis.” You should propose theoretical hypotheses based on existing literature and theories through reasoning. Instead of simply adding a sentence, we assume that..., without reasoning process and theoretical and empirical basis.
Response 4: Thanks a lot for your valuable comment. This paper explores the advantages and disadvantages of digital and analog clock display formats on the cognitive performance of integrated meteorological radar information through two experiments. In the first five paragraphs of the Introduction, we provide a literature review of previous relevant research, including reasoning process and theoretical and empirical basis. The key scientific questions that this study aims to explore are then presented in the sixth paragraph. In the seventh and eighth paragraphs, the main purpose of experiment 1 and experiment 2 and the corresponding experimental hypotheses are presented respectively. Although the layout of the article is somewhat different from what you mentioned, the Introduction dose include the reasoning process and theoretical empirical basis of the hypotheses.
Please refer to the red content in 1. Introduction section of the revised manuscript for specific modifications.
Round 3
Reviewer 3 Report
Comments and Suggestions for Authors
1. Write the supplementary experiment as Experiment 2 in the paper
2. Delete the "random recruitment" in all experiments
Comments on the Quality of English Language1. Write the supplementary experiment as Experiment 2 in the paper
2. Delete the "random recruitment" in all experiments